# Recommendation of Tahiti acid lime cultivars through Bayesian probability models

**Renan Garcia Malikouski**[1⊕], **Filipe Manoel Ferreira**[2⊕], **Saulo Fabrício da Silva Chaves**[3⊕], **Evellyn Giselly de Oliveira Couto**[3⊕], **Kaio Olimpio das Graças Dias**[1⊕], **Leonardo Lopes Bhering**[1⊕]*

**1** Departamento de Biologia Geral, Universidade Federal de Viçosa, Viçosa, Minas Gerais, Brazil, **2** Department of Crop Science—College of Agricultural Sciences, São Paulo State University, Botucatu, São Paulo, Brazil, **3** Departamento de Agronomia, Universidade Federal de Viçosa, Viçosa, Minas Gerais, Brazil

⊕ These authors contributed equally to this work.
* leonardo.bhering@ufv.br

## Abstract

Probabilistic models enhance breeding, especially for the Tahiti acid lime, a fruit essential to fresh markets and industry. These models identify superior and persistent individuals using probability theory, providing a measure of uncertainty that can aid the recommendation. The objective of our study was to evaluate the use of a Bayesian probabilistic model for the recommendation of superior and persistent genotypes of Tahiti acid lime evaluated in 12 harvests. Leveraging the Monte Carlo Hamiltonian sampling algorithm, we calculated the probability of superior performance (superior genotypic value), and the probability of superior stability (reduced variance of the genotype-by-harvests interaction) of each genotype. The probability of superior stability was compared to a measure of persistence estimated from genotypic values predicted using a frequentist model. Our results demonstrated the applicability and advantages of the Bayesian probabilistic model, yielding similar parameters to those of the frequentist model, while providing further information about the probabilities associated with genotype performance and stability. Genotypes G15, G4, G18, and G11 emerged as the most superior in performance, whereas G24, G7, G13, and G3 were identified as the most stable. This study highlights the usefulness of Bayesian probabilistic models in the fruit trees cultivars recommendation.

## Introduction

Breeding perennial fruit crops presents a set of challenges. The 3 to 5-year juvenile phase and the variable expression of quantitative traits over time can delay and mislead the selection of superior genotypes [1]. This differential performance over time can be a reflex of the genotypes-by-harvests interaction (GHI). The GHI in perennial species refers to the variation in gene expression and, consequently, the phenotypic traits of a plant due to the different environmental conditions and agricultural practices that occur in each planting cycle [2]. Therefore, in perennial fruit breeding, repeated measures on the same plant over time are important, which increases costs and the duration of breeding cycle [3]. In the presence of

**Data Availability Statement:** The data relevant to this study are available from GitHub at: https://github.com/malikouskirg/PONE-D-23-38484.

**Funding:** This study was financed in part by the Coordenação de Aperfeiçoamento de Pessoal de Nível Superior (CAPES) - Finance Code 001, Conselho Nacional de Desenvolvimento Científico e Tecnológico (CNPq), Fundação de Amparo à Pesquisa do Estado de Minas Gerais (FAPEMIG), Fundação de Amparo à Pesquisa do Estado do Espírito Santo (FAPES), and Fundação de Amparo à Pesquisa do Estado de São Paulo (FAPESP). FMF was supported by FAPESP (São Paulo Research Foundation, Grant 2023/04881-3), and LLB was supported by CNPq (Research Productivity Fellowship, Grant 310610/2021-4).

**Competing interests:** The authors have declared that no competing interests exist.

complex GHI, breeders must consider the productivity and the stability of the genotypes over harvests [4–6].

In Tahiti acid lime (*Citrus latifolia* Tanaka L.), several strategies have been proposed to address the challenges related to the extended juvenile phase and the presence of GHI. Grafting, a well-established technique in citrus propagation, influences vegetative growth and shortens the juvenile phase, facilitating earlier evaluations in breeding programs [7, 8]. Furthermore, research has demonstrated that employing repeatability models allows for accurate genotype selection in Tahiti acid lime after just four measurements [9]. To address the interdependence of measurements taken on the same individual over time, approaches like random regression models using Legendre polynomials have been applied [10]. Random regression models enable the estimation of the genotypic trajectory of evaluated treatments over time.

A recent methodology proposed by Dias et al. [11] can help to optimize the Tahiti acid lime cultivars recommendation, since it uses probability concepts. This method aims to reduce the risk associated with the selection of a given genotype, which is the daily dilemma of the farmers, who seek to guide their actions to minimize the risks of low production for a given crop [12]. Furthermore, plant breeding is increasingly focused on developing genotypes capable of coping with the modifications of the current climate, as the impacts of climate change become an additional pivotal factor to consider in the agricultural sector [13]. Dias et al. [11] proposed to use Bayesian probability concepts to assist in the selection of genotypes that gather favorable alleles for performance and stability across environments and harvesters. Furthermore, it allows a straightforward recommendation based on the probability of a given genotype to be selected considering its performance and stability, and a pairwise comparison of the probabilities of the evaluated selection candidates. This methodology has been proposed for the multiple-location context. Nevertheless, we believe that the same ideas are valid for the multi-harvest. The objective of our study was to evaluate the use of a Bayesian probabilistic model for the recommendation of superior and persistent genotypes of Tahiti acid lime evaluated in 12 harvests.

## Materials and methods

### Trial and plant material

We evaluated 24 combinations of rootstock and scion of Tahiti acid lime for fruit yield, expressed in kg of fruit per plant (Kg/tree). The hybrids *Citrumelo swingle* (*Citrus paradisi* X *Poncirus trifoliata*) and *Citrandarin 'riverside'* (*Citrus sunki* X *Poncirus trifoliata*) were used as rootstock for scions of 12 clones of Tahiti acid lime (Table 1). The plant materials came from the Active Germplasm Bank of Embrapa Mandioca e Fruticultura [14]. Each combination was considered a different selection candidate. The trial was laid out in a complete randomized block design, with four replications. Each plot was composed of three plants. The inter-rows and inter-ranges spacing was 6 x 3 m, respectively. We performed the statistical analysis described in the next topic using the plots unit mean.

The trial was established in July 2015, in São Mateus municipality, Espírito Santo, Brazil (18°48'21"S, 39°53'30"W, 35 m of altitude). The work was conducted on a farm at Bello Fruit® company, through a partnership between the fruit production and export company and the Universidade Federal do Espírito Santo. The experimental region has a rainy season in summer and a dry season in winter, being classified as Aw, following the classification of Köppen [15]. The precipitation and temperature during the period of the experiment was illustrated in the S1 Fig of the Supplementary Material. The data was collected between July 2017 and September 2020, and consisted of 12 harvests carried out in the following days after planting: 736, 808, 861, 918, 972, 1083, 1200, 1249, 1415, 1568, 1633 and 1867.

**Table 1. Codes for 24 combinations of rootstock and scion of Tahiti acid lime.**

| Scion | Rootstock | |
|---|---|---|
| | *Citrumelo swingle* | *Citrandarin riverside* |
| Bello Fruit | G1 | G13 |
| Eledio | G2 | G14 |
| Iconha | G3 | G15 |
| Itarana | G4 | G16 |
| Santa Rosa | G5 | G17 |
| Bearss Lime | G6 | G18 |
| CNPMF01 | G7 | G19 |
| CNPMF02 | G8 | G20 |
| CNPMF2001 | G9 | G21 |
| CNPMF5059 | G10 | G22 |
| BRS Passos | G11 | G23 |
| Persian 58 | G12 | G24 |

## Statistical analyses

We applied the probabilistic approach of Dias et al. [11] by fitting two Bayesian models using the the *rstan* package [16] and ProbBreed package [17]. The first model had a homogeneous residual variance (B-ID) and presented the following conditional normal probability:

$$y_{ijk} N \left( E\left[ y_{ijk} \right], \sigma \right)$$

where:

$$E\left[ y_{ijk} \right] = \mu + g_i + r_j + h_k + gh_{ik} + p_{ij} + e$$

in which $E[y_{ijk}]$ is the expectation of the phenotype from the $i^{th}$ genotype, evaluated in the $j^{th}$ block, at the $k^{th}$ harvest; $\mu$ is the overall mean; $g_i$ is the genotypic effect; $r_j$ is the block effect; $h_k$ is the harvest effect; $gh_{ik}$ is the genotypes-by-harvests interaction, and $p_{ij}$ is the environmental permanent effect.

The prior probability distribution of each parameter of the model was defined as:

$$\mu \sim N(0, \sigma_{[\mu]})$$

$$r \sim N(0, \sigma_{[r]})$$

$$h \sim N(0, \sigma_{[h]})$$

$$g \sim N(0, \sigma_{[g]})$$

$$gh \sim N(0, \sigma_{[\mu]})$$

$$p \sim N(0, \sigma_{[p]})$$

$$e \sim Half\ Cauchy(0, \sigma_{[e]})$$

$$\alpha \sim Half\ Cauchy(0, \sigma_{[\alpha]})$$

where $N(0,\sigma_{[\alpha]})$ and *Half Cauchy*$(0,\sigma_{[\alpha]})$ represent the normal and half-Cauchy distributions, respectively, with mean equal to zero and with different $\sigma^2_{[\alpha]}$ scale parameters. The following hyperpriors were considered for the respective parameters:

$$\sigma_{[\mu]} \sim Half\ Cauchy(0,\varphi)$$

$$\sigma_{[r]} \sim Half\ Cauchy(0,\varphi)$$

$$\sigma_{[h]} \sim Half\ Cauchy(0,\varphi)$$

$$\sigma_{[g]} \sim Half\ Cauchy(0,\varphi)$$

$$\sigma_{[gh]} \sim Half\ Cauchy(0,\varphi)$$

where $\varphi$ represents a predetermined global hyperparameter ($\varphi = max(y)*10$), defined in such a way that results in a weakly informative second level hyperpriors. Therefore, the data dominated the posterior distributions [18]. The half-Cauchy distribution is restricted to positive values, being often recommended as a prior distribution when modeling variance parameters [19].

The second Bayesian model had heterogeneous residual variances (B-DG). This model has the same considerations of the M-ID, except that $\sigma_k HalfCauchy(0,\sigma_{[\sigma k]})$. We selected the best-fitted model via the Watanabe-Akaike Information Criterion 2 (WAIC2) [18]. We used the Hamiltonian Monte Carlo algorithm in four Markov chains with 4000 samples, thin equals 1 and 50% burn-in.

**Convergence diagnostics.** The scale reduction factor ($\hat{R}$) was used to assess the effectiveness of the convergence of the Markov chain Monte Carlo (MCMC). This metric indicates whether the chains have mixed sufficiently, and the estimated model parameters have reached a stable distribution. The closer the $\hat{R}$ to 1, the greater the quality of mixing and convergence [18]. Greater values imply that more iterations are needed.

We also conducted a graphical analysis to visually assess how well the data generated by our model aligns with the true generative process of the observed data. This involved creating samples (referred to as "$y_{gen}$") from the fitted models using ancestral sampling from the conditional joint distribution and then plotting these samples against the observed data. Additionally, we employed posterior predictive p-values to gauge how closely the statistical measures (maximum, minimum, median, mean, and standard deviation) of the data generated by the fitted models resembled those of the observed data [18]. For instance, when considering the maximum statistic, we defined the Bayesian p-value as follows: $P_{max} = pr(T(y_{gen},\theta) \geq T(y,\theta)|y)$, where $T$ is the statistic test. The degree of similarity between the statistics derived from the generated data and those from the observed data increases as the Bayesian p-values approach 0.5.

**Probability of superior performance and genotypic stability.** The probability metrics for both performance and stability utilized in this study were proposed by Dias et al. [11] and implemented in the ProbBreed R package [17]. Aiming to select the top four genotypes, we sampled the posterior distribution of the marginal genotypic values given the observed phenotypic values. In each sample, we ranked these genotypes in descending order of posterior genotypic values ($g_i^s$). Then, we counted the number of events where a given genotype appeared in the subset of superior genotypes ($\Omega$), i.e., among the top four (selection proportion of 16%). The selection of 4 genotypes has been defined in a breeding program that aims to select multiple superior materials to be recommended across various regions. Moreover, diversifying

varieties on a single farm is important for the sustainability of cultivating Tahiti acid lime. The probability of superior performance of a genotype is given by the number of samples it appeared in divided by the total number of samples. In summary:

$$Pr\left(g_i^s \epsilon \Omega_s \mid y\right) = \frac{1}{s} \sum_{s=1}^{S} T\left(g_i^s \epsilon \Omega_s \mid y\right)$$

where $S(s = 1,2,\ldots,S)$ is the number of samples and $I\left(g_i^s \in \mid y\right)$ is an indicator variable that maps failure (0, if $g_i^s \notin$) or success (1, if $g_i^s \in$).

We calculated the stability across harvests based on the variance of the effect of GHI. Genotypes with lower GHI variance ($var[gh_{ik}]$) tend to be more stable, showing less fluctuation in their performance between harvests. One can draw a parallel of this metric and the frequentist persistence (see the next topic). Following the same idea described in the last paragraph, calculated the probability of a given candidate belonging to the subset of the top four genotypes with smaller $var[gh_{ik}]$ ($I(var[gh_{ik}]^s \in \ell V|y)$). The probability of superior stability was given as follows:

$$Pr(var(gh_i) \in \ell V \mid y) = \frac{1}{s} \sum_{s=1}^{S} I(var[gh_i]^s \in \ell V \mid y)$$

We used the ideas previously described for the two probabilities to perform pairwise comparisons between selection candidates. The goal is to investigate the chances of a given genotype being superior, whether in performance or stability, to its peers. The pairwise probabilities of superior performance and the pairwise probabilities of superior stability were given by, respectively:

$$Pr(g_i > g_i \mid y) = \frac{1}{S} \sum_{s=1}^{S} I\left(g_i^s > g_i^s \mid y\right)$$

and

$$Pr(var(gh_i) < var(gh_i) \mid y) = \frac{1}{s} \sum_{s=1}^{S} I(var[gh_i]^s < var[gh_i]^s \mid y)$$

where $I\left(g_i^S > g_i^S \mid y\right)$ is an indicator variable mapping success if $g_i^S$ has higher genotypic value than $g_i^S$, or failure otherwise; and $I(var[gh_i]^s < var[gh_i]^s|y)$ is another indicator variable mapping success if $gh_{ik}$ has lower variance than $gh_{ik}$, or failure otherwise.

The probability of superior performance within harvests and the pairwise probability of superior performance within harvests can be obtained by the following equations, respectively:

$$pr(g_{ik} > g_{ik} \mid y) = \frac{1}{S} \sum_{s=1}^{S} I\left(g_{ik}^S > g_{ik}^S \mid y\right),$$

$$pr(var(gh_{ik}) < var(gh_{ik}) \mid y) = \frac{1}{s} \sum_{s=1}^{S} I(var[gh_{ik}]^s < var[gh_{ik}]^s \mid y)$$

where $I\left(g_{ik}^s > g_{ik}^s \mid y\right)$, is an indicator variable mapping success if $g_i^S$ has higher genotypic value than $g_i^S$ in the harvest k, or failure otherwise; and $I(var[gh_{ik}]^s < var[gh_{ik}]^s|y)$ is another indicator variable mapping success if $gh_{ik}$ has lower variance than $gh_{ik}$ in harvest k, or failure otherwise.

**Stability in the frequentist context.** To compute the stability, we first fitted the following frequentist model (F-DG):

$$y = 1\mu + X_1 r + X_2 h + Z_1 g + Z_2 gh + Z_3 p + e$$

where $y$ is the vector of phenotypic data, 1 is a vector of ones, $\mu$ is the intercept of the model, $r$

is the vector of repetition effects (assumed to be fixed), $h$ is the vector of harvest effects (assumed to be fixed), $g$ is the vector of genotypic effects (assumed to be random) $\left[g \sim N\left(0, I\sigma_g^2\right)\right]$, $gh$ is the vector of the effects of the genotype-by-harvest interaction (assumed to be random) $\left[gh \sim N\left(0, I\sigma_{gh}^2\right)\right]$, $p$ is the vector of permanent environmental effect (assumed to be random) $\left[p \sim N\left(0, I\sigma_p^2\right)\right]$, and $e$ is the vector of residues associated with phenotypic observations (random) $[e \sim N(0, R)]$, where $R$ is a residual covariance matrix. The capital letters $X_1$ and $X_2$ refer to the incidence matrix for the fixed effects, and $Z_1, Z_2$ and $Z_3$ are the incidence matrix for the random effects of the respective effects.

We estimated the variance components and predicted the genetic values using the residual maximum likelihood–REML [20], and the best linear unbiased predictor–BLUP [21], respectively. The significance tests of the random effects were verified via likelihood ratio test (LRT) [22].

A concept analogous to stability called "persistence" is used by breeders of perennial forages and refers to the ability to survive and keep producing dry matter for long periods [23, 24]. In perennial fruit plants, this concept can be readjusted as the ability to maintain a high fruit yield for several years [2]. Therefore, persistence is analogous to an ecological stability of the genotypes ($P_i$) based on the distance between each genotype in relation to the ideotype. The ideotype ($g_{max}$) was defined as the maximum genotypic value estimated on each harvest. We used the following expression to estimate the persistence [24]:

$$P_i = \frac{\frac{1}{\sum_{k=1}^{12}(g_i - g_{max})^2}}{\sum_{i=1}^{24}\frac{1}{\sum_{k=1}^{12}(g_i - g_{max})^2}}$$

We compared the genotype ranking of the Bayesian probabilistic model and persistence via the frequentist model using Spearman correlation, following the expression below [25]:

$$\rho = 1 - \frac{6\sum_{i=1}^{n}d^2}{n(n^2 - 1)}$$

where $\rho$ is the Spearman correlation, $d$ is the difference between the rank positions of the genotypes in each methodology, and $n$ is the number of genotypes.

We performed the analysis using *R* software environment, version 4.2.1 [26]. The Bayesian models were fitted using the probabilistic programming language *Stan* [27] from the *rstan* package [16], and the ProbBreed R package [17]. We fitted the linear mixed models using *ASReml-R* (version 4.1) [28].

## Results

### Probability of superior performance of genotypes

The Bayesian models (B-ID and B-DG) displayed a mean value of the statistic $\hat{R}$ close to 1, indicating strong convergence of the model parameters (Table 2). Notably, among these models, the B-DG model exhibited the best fit, as evidenced by its lower WAIC2 value (Table 2). Note how the density of the data generated by B-DG model follows the same trend as the density of the real data. This indicates the model's effectiveness in replicating the distribution of observed data through the generated data, highlighting its reliability in capturing the underlying patterns (Fig 1A). Considering the findings from the B-DG, the posterior distribution of genotypic values of the 24 genotypes exhibited a variable overlapping pattern among their highest posterior density intervals (Fig 1B).

**Table 2. Comparative statistics of Bayesian models.**

| Predictive a posteriori verification statistic | Homogeneous residual variance model | Heterogeneous (diagonal) residual variance model |
|---|---|---|
| WAIC2 | 8420.87 | 6899.61 |
| $\hat{R}$ | 1.00 | 0.99 |

Where WAIC2 is the Watanabe–Akaike information criterion, and $\hat{R}$ is the potential scale reduction factor.

Genotypes G15, G4, G18 and G11 presented the highest posterior genotypic values (Fig 1B) and the highest probability of superior performance (Fig 1C). Genotypes G15, G4, G18, G11, G3, G23, G22, G1, G19, G14, G2, G13 and G16 were selected at least in a few samples, while the remaining 11 genotypes did not appear among the selected in any sample (Fig 1C). Genotypes G15 and G4 offers low risk of bad performance if selected (probability of superior performance equal to or higher than 75%). Genotypes G24 and G7 had the highest probability of superior stability, meaning that they have the less variable performance across harvests (Fig 1D).

The pairwise comparison graph presents two symmetrical sides, which indicate the probability of success (lower diagonal) and failure (upper diagonal) of the genotypes in the *x*-axis being superior to the ones of the *y*-axis (Fig 2A). G15, for example, has a high probability of beating all genotypes. On the other hand, G9 is beaten by all its peers, except for G21, which wins three-thirds of the time. The greenish color indicates genotypes that tie in their performance (probability close to 50%), like G18 and G11, G2 and G14, and G10 and G24 (Fig 2A).

The probabilities of a given genotype belonging to the group of selected ones in each harvest varies for high performance genotypes (Fig 2B). Genotypes G15, G4, G18, and G11 consistently displayed probabilities exceeding 50% across nearly all harvests. Conversely, genotypes G9 and G21 exhibited nil probabilities throughout all harvests, implying that they are not recommended due to their consistently poor performance (Fig 2B). According to the probability of superior performance within the harvests, high production by certain genotype in one harvest did not guarantee the same level of performance in subsequent harvests (Supplementary material–S2 Fig).

## Probability of genotypic stability

The probability of encountering a Tahiti acid lime genotype with minimal variation for stability was generally low. Only G24 exhibited values exceeding 0.3, indicating it to be the most stable genotype among those under evaluation (Fig 1D). The frequentist model (F-DG) demonstrated the significance of GHI according to the LRT. Also, the variance components exhibited similar magnitudes for both models (Table 3). The $\sigma_g^2$ overshadowed the $\sigma_{gh}^2$ and $\sigma_p^2$, with values of 2.21 for the F-DG model and 2.44 for the B-DG. The residual variance displayed varying values across different harvests, ranging from 0.88 to 412.17 in the F-DG and from 0,63 to 384.17 in the B-DG (Table 3).

Persistence by the frequentist sense presented different results from the probability of superior stability of Bayesian models. However, in both contexts, the values were low. Genotypic persistence in the frequentist context ranged from 5.7 to 3.4 (Fig 3). Except for G3, which presented the highest value and well above the others, the other genotype values were in a range of 1.1, showing the low ability of the F-DG to distinguish the persistence of these genotypes. The correlation between the persistence rankings in the Bayesian context (Fig 1D) and those in the frequentist context (Fig 3) exhibited a coefficient of 0.69. This suggests a statistically significant correlation between the classifications provided by the two methods, considering a confidence level of $\alpha = 0.05$.

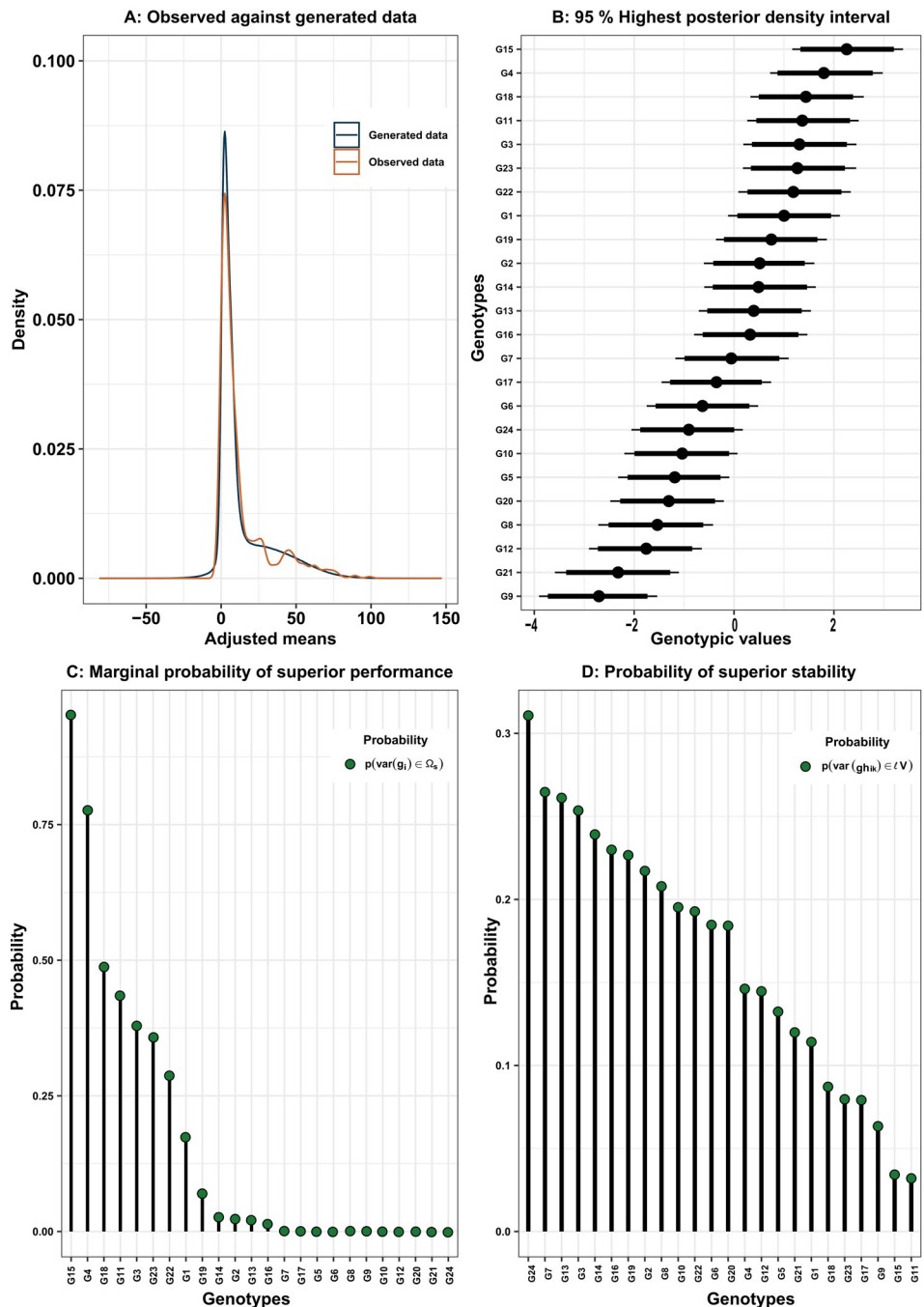

**Fig 1.** Bayesian distribution of the observed and generated data of the Tahiti acid lime dataset (A). Caterpillar plot of the genotypic posterior effects (and their 95% and 97.5% HPDs, represented by the thick and thin lines, respectively) of 24 genotypes of posterior effects (B). Marginal probability of superior performance of the 24 genotypes (C). Probability of superior stability of the 24 genotypes (D).

## Discussion

The consideration of the GHI is very important in the genetic evaluation of perennial species. This is because gene expression varies in response to environmental factors across different harvests [29]. The model selection based on WAIC2, coupled with the observed increase in

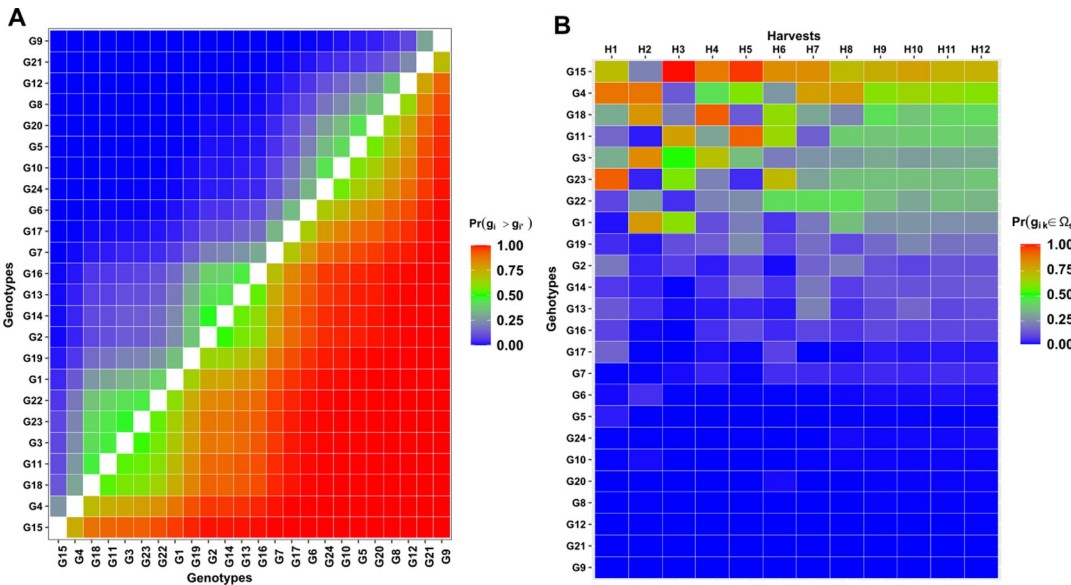

**Fig 2.** Pairwise probability of superior performance among genotypes (A). Probabilities of superior performance within environments (B).

residual variance over successive harvests, provides robust evidence in favor of the suitability of the heterogeneous residuals model for data fitting over the homoscedastic model. These findings underscore the critical importance of appropriate modeling and accounting for diverse sources of variation in repeated measures datasets.

**Table 3. Variance components estimates for a frequentist heterogeneous (diagonal) residual variance model (F-DG) and a Bayesian heterogeneous (diagonal) residual variance model (B-DG).** For the Bayesian model, it includes the corresponding lower (L) and upper (U) highest posterior density (HPD), considering a confidence level $\alpha = 0.05$.

| Parameter | | F-DG | | B-DG | |
|---|---|---|---|---|---|
| | | Component | L-HPD | Component | U-HPD |
| $\sigma_g^2$ | | 2.21 | 1.31 | 2.44 | 4.22 |
| $\sigma_{gh}^2$ | | 0.81 | 0.44 | 0.76 | 1.09 |
| $\sigma_p^2$ | | 0.24 | 0.11 | 0.26 | 0.42 |
| $\sigma_e^2$ | H1 | 1.19 | 0.89 | 1.24 | 1.67 |
| | H2 | 0.88 | 0.63 | 0.91 | 1.29 |
| | H3 | 0.86 | 0.70 | 0.97 | 1.33 |
| | H4 | 5.60 | 4.48 | 5.88 | 7.69 |
| | H5 | 3.07 | 2.72 | 3.60 | 4.72 |
| | H6 | 4.22 | 3.37 | 4.52 | 5.99 |
| | H7 | 10.26 | 8.53 | 11.23 | 14.78 |
| | H8 | 20.17 | 20.57 | 26.71 | 34.32 |
| | H9 | 412.17 | 328.66 | 418.67 | 531.69 |
| | H10 | 136.74 | 113.61 | 146.63 | 186.56 |
| | H11 | 404.45 | 324.73 | 413.08 | 521.54 |
| | H12 | 473.61 | 384.17 | 492.71 | 621.24 |

$\sigma_g^2$: genotypic variance, $\sigma_{gh}^2$: genotypes-by-harvesters interaction, $\sigma_p^2$: variance of the environmental permanent effect, and $\sigma_e^2$: residual variance for the 12 harvesters (H1 to H12).

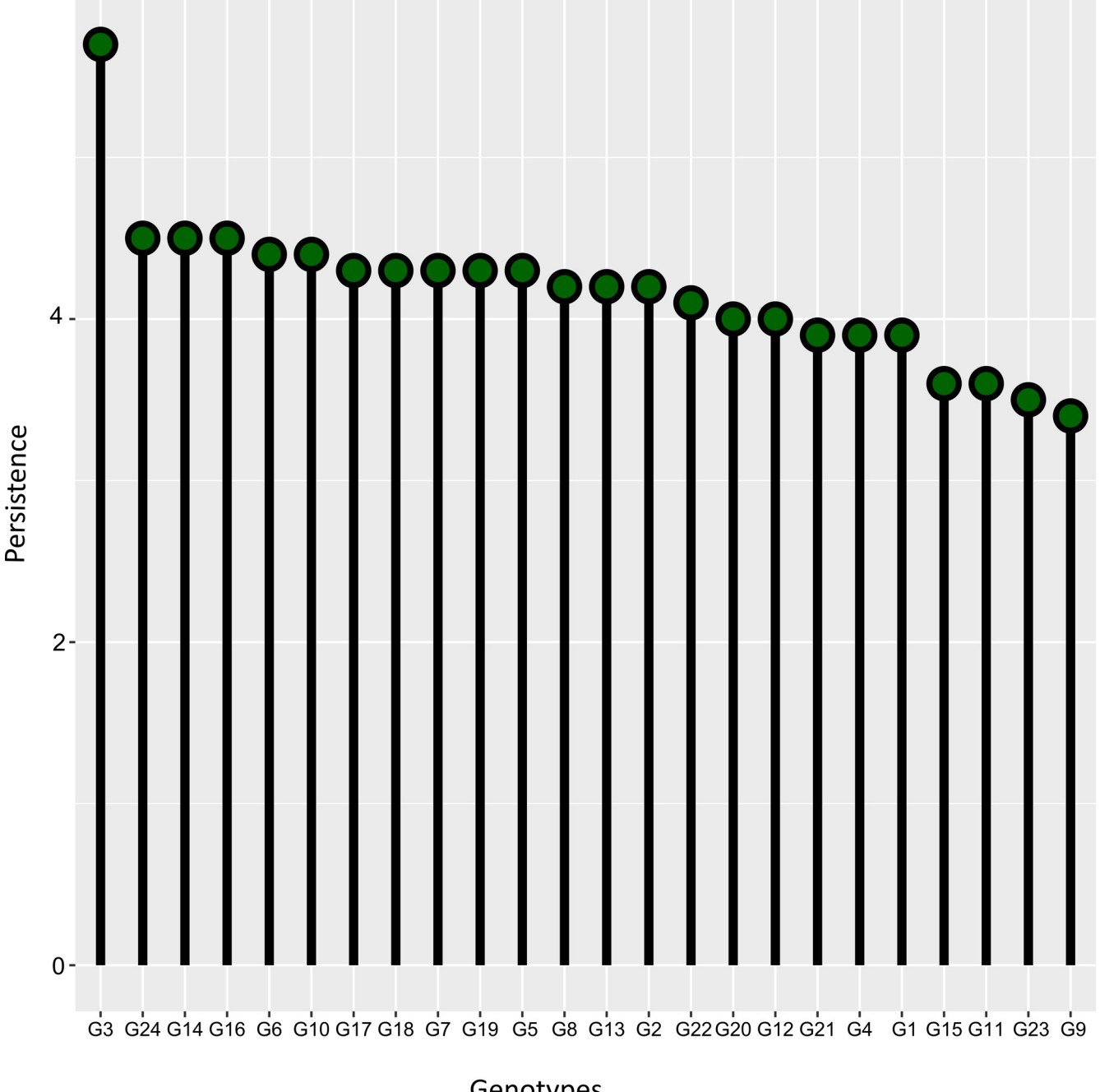

**Fig 3. Persistence of 24 genotypes of Tahiti acid lime via the frequentist model with diagonal residual variance.**

Comparing the genotypic values associated with a posteriori probability ensures greater confidence in the analysis of the performance of the Tahiti acid lime genotypes. Bayesian probability measures offer breeders the opportunity to delve into the probability of a particular genotype surpassing others, including scenarios where a candidate genotype may outperform a widely adopted cultivar [11]. This probability-based approach aids decision-making, especially when the difference between genotypes' performance is small. These probabilities are

dependent on the selection intensity, a value that is often predefined in breeding programs, depending on the stage [30]. Thence, probabilities are an intuitive metric and offer an enhanced reliability for recommendations, since it provides information about the risks. Its application simplifies decision-making processes and opens avenues for application in various domains beyond plant breeding [11]. Defining a probability value as a threshold would be very useful for practical purposes, as it would make it easy to classify comparisons as significative or non-significative. However, due to the different selective intensities that can be employed, and the diversity of selection candidates, the threshold depends on each reality and dataset the analysis is adjusted for.

Genotypes with the highest probability of superior performance, namely G15 (Iconha x Citrandarin riverside), G4 (Itarana x Citrumelo swingle), G18 (Bearss Lime x Citrandarin riverside), and G11 (BRS Passos x Citrumelo swingle), emerged as strong candidates for recommendation. These genotypes, as determined through probabilistic methods, possess alleles that impart adaptation to the changes in environmental conditions encountered through the harvests, maintaining consistently good performance. Indeed, the probability of superior performance is a measure of stability in an agronomic sense. The presence of GHI imposes changes for the selection of superior genotypes based on a single or a few harvests. Therefore, considering the genotype's performance across multiple harvests is advisable for making well-informed decisions when selecting genetically superior candidates.

Initially employed in forage species to assess the maintenance of productivity levels through multiple cuts [24], the concept of persistence is also relevant in the context of perennial fruit crops [31], given the perennial behavior of both. We can make a parallel between persistence and the probability of superior stability, as both represent ecological stability, i.e., invariance of performance. In the Bayesian framework, the probability of being selected among the four most stable genotypes was, in general, low (values below 0.4). This metric had a 50% agreement rate in identifying the top four most persistent genotypes with the frequentist persistence. Both approaches selected G24 and G3 among the four most persistent genotypes.

Certain advantages of the Bayesian model deserve attention. The incorporation of priors enhances confidence in selecting materials with varying levels of persistence [11, 32]. Furthermore, Bayesian models offer the advantage of obtaining variance components with associated high probability density intervals. These credibility intervals provide a more intuitive means of quantifying component uncertainty. Also, from an asymptotic perspective, Bayesian credibility intervals outperform frequentist confidence intervals [18], since frequentist confidence intervals may prove inaccurate for small or moderate sample sizes and may, in certain instances, fail to converge to the true parameter value as the sample size increases [33]. Likewise in mixed model, Bayesian models work well in common situation of plant breeding, such as unbalanced data, heterogeneous residual variance [6, 34, 35].

## Conclusion

By applying probabilistic Bayesian models in Tahiti acid lime in the genetic evaluation, we estimated the probability of superior performance of a genotype and the pairwise probabilities of superior performance between genotypes for both across and within harvests. Genotypes G15, G4, G18 and G11 were considered superior, and genotypes G24, G7, G13 and G3 were considered the most stable ones. Therefore, we believe that Bayesian probabilistic models can assist to more accurate recommendation in perennial fruit crops evaluated along many harvests, since it allows a more direct and precise interpretation of the performance and persistence of the candidate's genotypes.

## Supporting information

**S1 Fig. Climatic data of precipitation (mm) and temperature (˚C) from July 2017 to July 2020 in the field trial location.**
(TIF)

**S2 Fig. Pairwise probability of superior performance among genotypes within harvest.**
(TIF)

## Acknowledgments

We appreciate the Federal University of Espírito Santo, the Brazilian Agricultural Research Corporation, and the Bello Fruit Company for making possible the execution of these experiments. Also, we thank the Federal University of Viçosa, which provided infrastructure and human training for data analysis.

## Author Contributions

**Conceptualization:** Renan Garcia Malikouski, Kaio Olimpio das Graças Dias.

**Formal analysis:** Renan Garcia Malikouski, Filipe Manoel Ferreira, Saulo Fabrício da Silva Chaves, Kaio Olimpio das Graças Dias.

**Investigation:** Saulo Fabrício da Silva Chaves, Evellyn Giselly de Oliveira Couto.

**Methodology:** Saulo Fabrício da Silva Chaves.

**Supervision:** Evellyn Giselly de Oliveira Couto, Leonardo Lopes Bhering.

**Validation:** Evellyn Giselly de Oliveira Couto, Leonardo Lopes Bhering.

**Writing – original draft:** Renan Garcia Malikouski, Evellyn Giselly de Oliveira Couto, Kaio Olimpio das Graças Dias.

**Writing – review & editing:** Filipe Manoel Ferreira.

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
