## [Decision Letter · Decision Letter 0]

26 Dec 2023

PONE-D-23-38484Application of Bayesian probabilistic models for recommendation of ‘Tahiti’ acid lime using longitudinal dataPLOS ONE

Dear Dr. Bhering,

Thank you for submitting your manuscript to PLOS ONE. After careful consideration, we feel that it has merit but does not fully meet PLOS ONE’s publication criteria as it currently stands. Therefore, we invite you to submit a revised version of the manuscript that addresses the points raised during the review process.

We look forward to receiving your revised manuscript.

Kind regards,

Mehdi Rahimi, Ph.D.

Academic Editor

PLOS ONE

Journal Requirements:

4. Thank you for stating the following in the Acknowledgments Section of your manuscript: "The authors are grateful to the Coordenação de Aperfeiçoamento de Pessoal de Nível Superior (Capes, Code 001), Conselho Nacional de Desenvolvimento Científico e Tecnológico (CNPq), Fundação de Amparo à Pesquisa do Estado de Minas Gerais (Fapemig) and Fundação de Amparo à Pesquisa do Estado do Espírito Santo (Fapes). Filipe Manoel Ferreira was supported by FAPESP (São Paulo Research Foundation, Grant 2023/04881-3),"

Please remove any funding-related text from the manuscript and let us know how you would like to update your Funding Statement. Currently, your Funding Statement reads as follows: "“The authors received no specific funding for this work.”

Reviewers' comments:

Reviewer's Responses to Questions

**Comments to the Author**

1. Is the manuscript technically sound, and do the data support the conclusions?

Reviewer #1: Yes

Reviewer #2: Yes

2. Has the statistical analysis been performed appropriately and rigorously? 

Reviewer #1: Yes

Reviewer #2: Yes

3. Have the authors made all data underlying the findings in their manuscript fully available?

Reviewer #1: No

Reviewer #2: Yes

4. Is the manuscript presented in an intelligible fashion and written in standard English?

Reviewer #1: Yes

Reviewer #2: Yes

5. Review Comments to the Author

Reviewer #1: The authors aimed to use probability concepts to recommend Tahiti acid lime genotypes. To do this, they evaluated, in a field trial, 24 combinations of rootstock and scion of Tahiti acid lime regarding fruit yield, over 12 harvests. They concluded that the Bayesian analysis outperformed the frequentist analysis. This work appears to be the first to use probability concepts to recommend cultivars of fruit trees and has merit for publication in PLOS ONE.

Random regression models are the state-of-the-art for the analysis of repeated measures. They are parsimonious and deal with genotype-harvest interactions. In this work, Bayesian and frequentist random regression models could provide better results?

See more comments in the attached file.

Reviewer #2: In my point of view, the article is comprehensive in terms of statistical analyses and arguments that support its hypotheses. The writing is well-structured, lacking only some additional information about the cultivation of the 'Tahiti' acid lime, which would further reinforce the importance of the article. Therefore, my recommendation is to accept it for publication with minor revisions.

6. PLOS authors have the option to publish the peer review history of their article (what does this mean?). If published, this will include your full peer review and any attached files.

Reviewer #1: No

Reviewer #2: No

---

## [Author Response · Author response to Decision Letter 0]

27 Jan 2024

To the Editorial Office of PLOS ONE

Dear Academic Editor Mehdi Rahimi, Ph.D.

We are sending the revised version of the manuscript PONE-D-23-38484. The changes based on the reviewers’ comments, were highlighted in blue in the manuscript. 

Each observation from the reviewers and the editor has been addressed in the file "Response to Reviewers."

Below, we paste the content present in the "Response to Reviewers" file.

To the Editorial Office of PLOS ONE

Dear Academic Editor Mehdi Rahimi, Ph.D.

We are sending the revised version of the manuscript PONE-D-23-38484. The changes, based on the reviewers’ comments, were highlighted in the reviewed manuscript file in blue color. We would like to thank the reviewers for the excellent contributions to the improvement of this manuscript. Below we responded individually to each comment made in the decision letter and reviewed manuscript. We are available for any further questions.

Academic Editor 

The authors are grateful for this observation from the Editor. A second check on the provided links was conducted, and a few changes were made, e.g., titles of captions and figures were bolded, and minor alterations were made to the acknowledgments. Other than that, everything is in accordance with the established norms. Changes are highlighted in blue in the Manuscript.

Thank you for this observation. We have added more information about the location of the experiment. This partnership does not present any conflict of interest, and there is no formal documentation regarding it. The company provides the site for carrying out the experiment and labor for managing the plants. In return, the university can generate fruit cultivation information for all the farmers in the region, without any right to exclusivity, and the transference of information takes place through the execution of field days held on the Bello Fruit farm.

Thank you and sorry for this misunderstanding. We have revised the Acknowledgments section, removing any issues of funding. However, we reiterate the need to disclose this information as funding agencies commonly require acknowledgment of their contributions in papers and abstracts. We added the information previously contained in the Acknowledgments to the cover letter (file named “cover_letter_financial_disclosure_changes.pdf”) so that it may be placed in the appropriate section on the published paper.

4. Thank you for stating the following in the Acknowledgments Section of your manuscript: "The authors are grateful to the Coordenação de Aperfeiçoamento de Pessoal de Nível Superior (Capes, Code 001), Conselho Nacional de Desenvolvimento Científico e Tecnológico (CNPq), Fundação de Amparo à Pesquisa do Estado de Minas Gerais (Fapemig) and Fundação de Amparo à Pesquisa do Estado do Espírito Santo (Fapes). Filipe Manoel Ferreira was supported by FAPESP (São Paulo Research Foundation, Grant 2023/04881-3),"

Please remove any funding-related text from the manuscript and let us know how you would like to update your Funding Statement. Currently, your Funding Statement reads as follows: "“The authors received no specific funding for this work.”

Thank you for the observation. As this point is related to the previous matter, it has already been addressed with the modifications mentioned in topic 3. We have modified the Acknowledgments section and have moved the paragraph mentioning the Funding information to the cover letter (file named “cover_letter_financial_disclosure_changes.pdf”) so that it may be placed in the appropriate section on the published paper.

We appreciate the comment and have now added the complete dataset that was used for the analyses to GitHub repository with public access. Consequently, we have inserted the link in the Supporting Information section. All authors of the manuscript have agreed to publish the full dataset to the scientific community. The link to access the dataset is:

https://github.com/malikouskirg/PONE-D-23-38484

Thank you for the information; we have reviewed the reference list, and everything is in accordance with the standards.

Reviewer #1

• Optimization of Tahiti Acid Lime Cultivars Recommendation Through Probability Concepts

Thank you for your observation. We believe that the name suggestion given by the reviewer is very relevant. We like the idea of adding the term "Bayesian" term to the title. We made a small adjustment to the reviewer’s idea. Thus, the new title is " Recommendation of ‘Tahiti’ acid lime cultivars through Bayesian probability models".

• Tahiti Acid Lime Breeding

Thank you for the suggestion. We adopted this short title. 

• any breeding program

We made the recommend. Thank you.

• a

Thank you, we add the letter “a” in the sentence “The objective of our study was to evaluate the use of a Bayesian probabilistic model”.

• The objective of our study was to evaluate the use of Bayesian probabilistic models for the recommendation of superior and persistent genotypes of ‘Tahiti’ acid lime evaluated in 12 harvests.

Thank you for your suggestion. We have removed the "s" from the word "models." However, we believe that keeping the word "persistent" would be important, as in addition to performance, the objective of this work was also focused on persistence.

• Genotypes G15, G4, G18, and G11 emerged as the most superior in performance, whereas G24, G7, G13, and G3 were identified as the most stable.

Thank you for your suggestion. However, we believe that this information is very important in practical terms, as it concerns the recommendation of genotypes. We have provided details about the coding in the Methods section, and therefore, the authors feel it is better to retain this information.

• Usefulness

Thank you, we changed for the suggested word. 

• the fruit trees cultivars recommendation

We made the recommend change. Thank you.

• In the presence of complex GHI, breeders must consider the productivity and the stability of the genotypes over harvests [3–5].

We made the recommend change. Thank you.

• To address the common issue of non-orthogonal longitudinal data, resulting from the interdependence…. Despite considerable progress in 'Tahiti' acid

lime breeding, straightforward methodologies for exploring GHI and identifying high performance and stable genotypes are yet to be established.

Thank you for the suggestion. We removed these parts of the text.

• optimize the Tahiti acid lime cultivars recommendation, since it uses probability concepts.

We made the recommend change. Thank you.

• Dias et al. [10] proposed to use probability concepts from Bayesian models to assist in the selection of genotypes that gather favorable alleles for performance and stability across environments.

We appreciate your suggestion and have adopted it in part, as retaining the information on alleles favorable for performance and stability is relevant. This is the genetic definition of the GHI interaction; thus, the text has been revised accordingly.

“Dias et al. [11] proposed to use Bayesian probability concepts to assist in the selection of genotypes that gather favorable alleles for performance and stability across environments and harvesters.”

• This methodology has been proposed for the multiple location context. Nevertheless, we believe that the same ideas are valid for the multi-harvest scenario

Thank you for the suggestion. We removed these parts of the text.

• The objective of our study was to evaluate the use of a Bayesian probabilistic model for the recommendation of superior and persistent genotypes of

Tahiti acid lime evaluated in 12 harvests.

Thank you. We have adopted your suggested objective.

• statistical

Thank you. We added this part to the manuscript.

• was

We made the recommend change. Thank you.

• Codes

We made the recommend change. Thank you.

• method

We made the recommend change. Thank you.

• How the permanent environmental effect was accounted here? Based on the results presented in Table 3, it seems that this effect was fitted, however it was not described here.

We appreciate the comment and apologize for the oversight. Indeed, there is a permanent environmental effect, which we have now correctly included in the model. With this adjustment, we identified values in the table that were switched. 

• N(0, σ[α])? N(0, σ[.]), where [.] can be μ, r, ..., e.

We revised the notations.

• We used a Monte Carlo Hamiltonian algorithm (the Hamiltonian Monte Carlo algorithm) in four Markov chains with 4000 samples (thin?) and 50% burn-in.

Thank you for the observation; we have revised the text to "the Hamiltonian Monte Carlo algorithm" and have added the thinning information of the model, which in this case was the default of the "rstan" package, 1.

• Package

Thank you. We changed the word “software” to “package”.

• Proportion

Thank you. We changed the word “intensity” to “proportion”.

• Effects

Thank you. We changed the word “effect” to “effects”.

• Was.. Were

Thank you. We changed the words “is/are” to “was/were”.

• Comparative statistics of Bayesian models using the 'Tahiti' acid file dataset.

Thank you for the suggestion; we have removed the term "using the Tahiti acid file dataset" from all captions, table titles, and figures where this text appeared.

• ‘Tahiti’

Thank you for your observation. In scientific articles, the word "Tahiti" is found both with and without quotation marks. Following the reviewer's suggestion, and after grammatical research, we decided to adopt the reviewer's advice to remove the quotation marks from the name Tahiti, resulting in 'Tahiti lime'. We have carried out the removal throughout the text.

• is the genotypes-by-harvesters interaction… is the variance of the environmental permanent effect

Thank you for the suggestion. We have changed the caption of Table 3 to the form proposed by the reviewer.

• by

Thank you. We changed the word “in” to “by”.

• important

Thank you. We changed the word “informative” to “important”.

• genetic evaluation

Thank you. We changed the word “assessment” to “genetic evaluation”.

• with heterogeneous residuals

Thank you. We changed the word “heterogeneous model” to “heterogeneous residuals model”.

• repeated measures

Thank you. We changed the word “longitudinal” to “repeated measures”.

• [https://doi.org/10.1007/s11295-023-01596-9]

Thank you. We have added this citation in the mentioned part of the text, as well as in the references.

• By applying probabilistic Bayesian models in longitudinal `Tahiti` acid lime dataset

Thank you. We have removed the terms "longitudinal" and “dataset” from the conclusion, as well as the previously mentioned quotation marks from 'Tahiti' throughout the text.

• This information is irrelevant to the reader.

As previously mentioned, the authors believe that maintaining the information about the recommended genotypes regarding performance and persistence is important in practical terms, since these materials are publicly available to farmers. This work, in addition to its statistical nature, is also aimed at assisting in the recommendation for planting of the Tahiti acid lime.

Reviewer #2

• I suggest adding brief information about the importance of the Tahitian lime culture, emphasizing the significance of work.

Thank you for your suggestion. Indeed, despite the statistical nature of the work, the authors believe that the importance of the crop should be highlighted in the introductory parts of the article, therefore, we have emphasized the importance of the culture in the initial sentences of the Abstract section.

• Highlight information about the juvenile phase of the culture. For example, how long does it last?

Thank you for the suggestion. The juvenile phase of the Tahiti acid lime can take from 3 to 5 years; we have added this information to the aforementioned sentence, highlighted in blue in the reviewed manuscript.

• Highlight to a non-expert what GHI means.

The authors thank you for your suggestion. We have added an introductory sentence that briefly defines the GHI interaction, with citation.

“The GHI in perennial species refers to the variation in gene expression and, consequently, the phenotypic traits of a plant due to the different environmental conditions and agricultural practices that occur in each planting cycle [2]”

• I wouldn't say they are mandatory, just that they are important for the recommendation and selection of superior individuals.

Thank you. We changed the word “mandatory” to “important”.

• I believe that studies have already been conducted exploring the performance and stability of genotypes. In this case, what is not yet present in the literature are works related to providing probabilities associated with these metrics. I suggest modifying this part.

We appreciate your suggestion. As a suggestion from the previous reviewer, we have removed that part of the manuscript.

• In a specific period of time or under certain adverse conditions... I suggest adding the part about climate challenges. Given the current variations in temperature and precipitation, it is necessary to select genotypes that can cope well with these uncertainties.

Thank you. Indeed, adding that part about climate change and its influence on breeding is very enriching for the work, thus we have included it in the introduction with a citation.

• and harvesters (measurements)

Thank you. We added this part in the text.

• plot unit mean

Thank you. We added this part in the text.

• ranges.

Thank you. We changed the word “columns” to “ranges”.

• Explain in more detail the reasons for having four selected genotypes

Thank you for your comment. The choice of 4 genotypes as selected was determined by the authors due to the fact that in a breeding program, there are typically several materials recommended, and not just one. Moreover, on the same farm, it's sometimes important to have a diversity of cultivars due to the susceptibility of some to biotic or abiotic factors. We have briefly added this explanation in the methods section.

• I believe that adding the calculation of the ratio between events with lower GHI and the total number of events would be interesting, as mentioned for performance.

We appreciate the suggestion. We have modified the text to make it similar to the definition of probability for performance. Below is the part added to the revised manuscript.

“Following the same idea described in the last paragraph, calculated the probability of a given candidate belonging to the subset of the top four genotypes with smaller ...”

• Explain in more detail the reason why knowing the genotypes in each harvest is important.

The authors appreciate your suggestion. However, we were unable to include that part in the materials and methods, as we believe that throughout the manuscript we aim to show that the objective of the work is performance and stability. In this case, evaluating performance at each separate harvest is a form of stability assessment, which is highlighted throughout the manuscript.

• Describe the Spearman correlation model as well as the citation.

We appreciate the suggestion. We have added the Spearman correlation estimator, as well as the original citation of the work.

• Indicate what is the probability value that this range varied (i.e., present the minimum and maximum values).

Thank you for your suggestion. However, since the figure does not show the exact frequentist persistence value, it is not possible to extract the minimum and maximum information from it. Nonetheless, the information on the range gives an idea of magnitude, and it is only from this that one can see that the variation was not so pronounced for the mentioned genotypes.

• In the materials and methods section, describe the correlation model and also cite relevant literature.

Thank you. As mentioned in a previous topic, we have added the estimator, as well as the reference for the Spearman correlation.

• Additionally, state that the variation in environmental conditions throughout different harvests creates diverse environments.

Thank you for this comment. 

• It would be interesting to indicate in this part of the text the correlation of this case with the existence of GHI.

Thank you for the observation, the correlation part was conducted between frequentist and Bayesian approaches; in this case, the authors believe that its linkage should be made further down in the discussion, and it was done, as you can see in the text below.

“In the Bayesian framework, the probability of being selected among the four most stable genotypes was, in general, low (values below 0.4). This metric had a 50% agreement rate in identifying the top four most persistent genotypes with the frequentist persistence. Both approaches selected G24 and G3 among the four most persistent genotypes.”

• It would be beneficial to indicate a probability value as thresholds. However, since it is still a new methodology, there may not be a well-defined value yet.

Thank you for your suggestion. Although it depends on various factors, having a threshold would be interesting. We have added the comment below in the revised manuscript.

“Defining a probability value as a threshold would be very useful for practical purposes, as it would make it easy to classify comparisons as significative or non-significative. However, due to the different selective intensities that can be employed, and the diversity of selection candidates, the threshold depends on each reality and dataset the analysis is adjusted for.”

• Certainly! It would be helpful to include the names of the genotypes here for the convenience of the readers. The format could be "GENOTYPE NAME (ENCODING)".

Thank you for your suggestion. As we defined the names of the genotypes by encoding throughout the manuscript, we changed to the suggested order, we put ENCODING (scion x rootstock).

• Because, in this case, both forage crops and fruit trees are perennials.

We appreciate your comment. We added the term "given the perennial behavior of both" to the mentioned sentence in the text.

---

## [Decision Letter · Decision Letter 1]

8 Feb 2024

Recommendation of Tahiti acid lime cultivars through Bayesian probability models

PONE-D-23-38484R1

Dear Dr. Bhering,

We’re pleased to inform you that your manuscript has been judged scientifically suitable for publication and will be formally accepted for publication once it meets all outstanding technical requirements.

Kind regards,

Mehdi Rahimi, Ph.D.

Academic Editor

PLOS ONE

Additional Editor Comments (optional):

Reviewers' comments:

Reviewer's Responses to Questions

**Comments to the Author**

1. If the authors have adequately addressed your comments raised in a previous round of review and you feel that this manuscript is now acceptable for publication, you may indicate that here to bypass the “Comments to the Author” section, enter your conflict of interest statement in the “Confidential to Editor” section, and submit your "Accept" recommendation.

Reviewer #1: All comments have been addressed

Reviewer #2: All comments have been addressed

2. Is the manuscript technically sound, and do the data support the conclusions?

Reviewer #1: Yes

Reviewer #2: Yes

3. Has the statistical analysis been performed appropriately and rigorously? 

Reviewer #1: Yes

Reviewer #2: Yes

4. Have the authors made all data underlying the findings in their manuscript fully available?

Reviewer #1: No

Reviewer #2: Yes

5. Is the manuscript presented in an intelligible fashion and written in standard English?

Reviewer #1: Yes

Reviewer #2: Yes

6. Review Comments to the Author

Reviewer #1: The manuscript is well organized and easy to follow. All issues in my original review were addressed.

Reviewer #2: I had already reviewed this manuscript with a recommendation of Minor Revision. The authors addressed the main issues raised by me and modified the manuscript accordingly. Therefore, my recommendation is to accept it for publication.

7. PLOS authors have the option to publish the peer review history of their article (what does this mean?). If published, this will include your full peer review and any attached files.

Reviewer #1: No

Reviewer #2: No

---

## [Editor Report · Acceptance letter]

22 Feb 2024

PONE-D-23-38484R1 

PLOS ONE

Dear Dr. Bhering, 

I'm pleased to inform you that your manuscript has been deemed suitable for publication in PLOS ONE. Congratulations! Your manuscript is now being handed over to our production team.

Kind regards, 

on behalf of

Associate Prof. Mehdi Rahimi 

Academic Editor

PLOS ONE